# Path Planning and Navigation of Miniature Serpentine Robot for Bronchoscopy Application

**DOI:** 10.3390/mi14050969

**Published:** 2023-04-28

**Authors:** Cheng-Peng Kuan, Shu Huang, Hao-Yan Wu, An-Peng Wang, Chien-Yu Wu

**Affiliations:** Mechanical and Mechatronics Systems Research Laboratories, Industrial Technology Research Institute, Hsinchu 31040, Taiwan; shu.huang@itri.org.tw (S.H.); apwang@itri.org.tw (A.-P.W.);

**Keywords:** miniature serpentine robot, robotic bronchoscopy, path planning and navigation

## Abstract

The miniature serpentine robot can be applied to NOTES (Natural Orifice Transluminal Endoscopic Surgery). In this paper, a bronchoscopy application is addressed. This paper describes the basic mechanical design and control scheme of this miniature serpentine robotic bronchoscopy. In addition, off-line backward path planning and real-time and in situ forward navigation in this miniature serpentine robot are discussed. The proposed backward-path-planning algorithm utilizes the 3D model of a bronchial tree constructed from the synthetization of medical images such as images from CT (Computed Tomography), MRI (Magnetic Resonance Imaging), or X-ray, to define a series of nodes/events backward from the destination, for example, the lesion, to the original starting point, for example, the oral cavity. Accordingly, forward navigation is designed to make sure this series of nodes/events shall be passed/occur from the origin to the destination. This combination of backward-path planning and forward navigation does not require accurate positioning information of the tip of the miniature serpentine robot, which is where the CMOS bronchoscope is located. Collaboratively, a virtual force is introduced to maintain the tip of the miniature serpentine robot at the center of the bronchi. Results show that this method of path planning and navigation of the miniature serpentine robot for bronchoscopy applications works.

## 1. Introduction

In the years since 2020, the world has been experiencing the impact of the COVID-19 pandemic. According to global statistics from the World Health Organization, more than 750 million people have been confirmed infected, and this number of infected continues to increase. Although the symptoms caused by the COVID-19 infection can be reduced via vaccinations, there still exists uncured injuries located deep within the lungs, such as pulmonary fibrosis. In addition, COPD (Chronic Obstructive Pulmonary Disease) causes 1 death every 10 s globally. COPD is caused by long-term inflammation in the respiratory tract, resulting in the airflow via the respiratory tract being strongly obstructed (obstructive ventilatory dysfunction). Essentially, COPD includes Chronic Bronchitis and Emphysema, of which the symptoms are irreversible and cannot be cured.

As described above, the symptoms (pulmonary fibrosis and obstructive ventilatory dysfunction) caused by the COVID-19 infection and COPD cannot be cured because of the irreversible injuries located deep within the lungs and bronchi. Researchers have tried to utilize Mesenchymal Stem Cells, which have the abilities of self-regeneration and multilineage differentiation, to cure irreversible diseases [1,2,3,4]. Golchin, A. et al. [1] showed the effectiveness of MSC therapy on COVID-19 based on MSCs’ immunomodulatory and regenerative properties and concluded that cost-effective utilization of MSC therapy is a critical issue. Kim, K. et al. [2] demonstrated that an intravenous injection of MSC is safe and acceptable. Mallis, P. et al. [3] concluded that MSCs can successfully be activated by a COVID-19 patient’s serum and secrete anti-inflammatory cytokines and growth factors. P.M. George et al. [4] also showed the potential of MSC therapy for COVID-19. 

On the other hand, Zhang, Y. et al. [5] adopted allogeneic adipose-derived mesenchymal stem cells (AMSCs) for COPD mice via intratracheal and intravenous administration. Their results [5] showed that MSC therapy for COPD is effective. In addition, ref. [5] showed that intratracheal administration of AMSCs is more effective compared to intravenous administration of AMSCs. Nejaddehbashi, F. et al. [6] also concluded that the anti-inflammatory effects of intratracheal AMSCs treatment are more potent than systemic administration for the treatment of emphysema. 

Based on previous studies [1,2,3,4,5,6], adopting MSC therapy for the treatment of COVID-19 and COPD may be a potential treatment solution, while a critical issue will be how to effectively deliver MSCs directly to the lesion. Therefore, research on MSC delivery is emerging. Go, G. et al. [7] designed microrobots with an overall scaffold size of 357.55 ± 18.57 μm and pore sizes of 43.85 ± 13.39 μm, which can contain MSCs. These microrobots containing MSCs can be injected via a catheter needle into the knee for cartilage regeneration in vivo and be navigated by an Electro-Magnetic Actuation (EMA) system [7]. This mechanism cannot be adopted in the lungs for delivering MSCs to the lesion because penetration via the skin by a catheter needle to the lungs has a very high probability of causing risky pneumothorax. 

Another resolution for delivering MSCs into the deep lungs (bronchi) could be a bronchoscopy via the human oral cavity. This kind of method depends heavily on the bronchoscopy operation experiences of the surgeons. Thus, studies and developments on robotic bronchoscopy have been conducted. Chen, A.C. et al. [8] showed the capability of a robotic bronchoscopy via the support of real-time images from a bronchoscope, electromagnetic navigation, and CT views. However, it still requires surgeons to control the movement of this robot-assisted bronchoscopy. This kind of serpentine robot [9,10,11,12] has been discussed as a feasible solution for moving inside the bronchi of the lungs.

Beyond the research on robotic bronchoscopy, path planning and navigation for this robotic bronchoscopy is a critical issue. Refs. [13,14] discussed several schemes to fulfill the functionalities of path planning and navigation. In [13], Ho, E. et al. reviewed several research studies related to path planning for a robotic bronchoscopy; for example, a bronchoscopist begins by reviewing the CT scan of the chest carefully and then writing down the plans to take—starting from the distal trachea to the main bronchus, lobar bronchus, segmental airway, and subsegmental airways, and finally the target. Another planning scheme mentioned in [13] is a computer-generated pathway where the bronchoscopist identifies and marks the target lesion and then the computer accordingly generates a pathway from the central airway to the target lesion. Regarding the navigation phase—guiding the robotics bronchoscopy while moving inside the bronchi—ref. [13] indicated that a synchronization procedure (Registration) between the position of the robotic bronchoscopy and the planned virtual pathway is required in advance. Without this synchronization, the robotic bronchoscopy could be navigated along the wrong path. Furthermore, ref. [13] described the significance of a live bronchoscopy view in order to correct the real-time movement of the robotic bronchoscopy. Duan, X., et al. [14] adopted an electromagnetic (EM) localization system integrated into the robotic bronchoscopy for further navigation.

After reviewing the research described above, we can draw the following conclusions: 1. MSC therapy for healing COVID-19 and COPD could be effective but requires a direct way to deliver the MSCs to the target in vivo; 2. adopting a miniature serpentine robot as a container for MSCs is an effective method of delivering MSCs directly to the target bronchi; accordingly, 3. path-planning and navigation inside the lungs become critical issues when an automatic robotic bronchoscopy is applied. Therefore, in this paper, a miniature serpentine robotic bronchoscopy is proposed. Section 2 of this paper will describe the basic mechanical design and control scheme of this proposed miniature serpentine robotic bronchoscopy. In addition, off-line backward path planning and real-time and in situ forward navigation of this miniature serpentine robot is discussed in Section 2. The implementation and results are described in Section 3. Then, Section 4 contains the discussion and conclusions. 

## 2. Mechanical Design, Control Scheme, Path Planning, and Navigation of the Miniature Serpentine Robotic Bronchoscopy

In this section, the mechanical design and control scheme of the proposed miniature serpentine robotic bronchoscopy will be described. A bending section with a length of 40 mm and an outer diameter of <4 mm is designed at the front portion of this robotic bronchoscopy for omnidirectional entry into the complicated bifurcation of bronchi. Then, a control scheme combining real-time visual information captured from a CMOS camera equipped at the tip of the bending section is specified. In addition, the algorithm for path planning and navigation of this robotic bronchoscopy is proposed. 

### 2.1. Mechanical Design

The main use case for this proposed robotics bronchoscopy is MSC therapy, consisting of delivering MSCs directly to the target lesion located deep within the bronchi. As is well known, the stricture of the bronchial tree is complicated with large turning angles (for example, 90°) and multiple bifurcations. Furthermore, the diameter of the bronchi becomes much smaller when moving deeper inside the bronchial tree. Accordingly, the outer diameter of this proposed robotic bronchoscopy must be as small as possible to allow the bronchoscopy to move closer to the lesion. In addition, real-time and in situ visual information is captured by a CMOS sensor/camera equipped at the tip of the bending section at the front portion of the proposed robotic bronchoscopy. To summarize, the basic design guidelines and requirements are listed in Table 1.

Based on Table 1, the mechanical structure of this proposed robotic bronchoscopy is shown in Figure 1, based on our previous works [15,16]. In Figure 1, a bending section at the front portion of the bronchoscopy is illustrated. At the tip of this bending section, there is a CMOS sensor/camera located. This bending section is manipulated by wires (tendons). Dimensions related to the mechanical structure are depicted in Figure 1 as well. The bending section is composed of several sub-sections with a 30° gap between two consecutive sub-sections, as shown in Figure 2. By accumulating several sub-sections to form a complete bending section, the maximal bending angle can be determined, and the radius of curvature/bending can be calculated.

For four movable directions—up–down and right–left—two sub-sections are connected via two hinges attached to each sub-section, interlaced mutually at 90° as shown in Figure 2. Accordingly, eight sets of two sub-sections can determine >180° bending angles along four directions, as illustrated in Figure 3. 

For further investigation and control of this mechanical design, kinematics analysis is required. To analyze the kinematics for this bending section, the well-known Denavit–Hartenberg (D-H) parameters were applied, and the D-H table of this bending section is shown in Table 2 [15,16], while the coordinates assigned to each sub-section are displayed in Figure 4 where the directions of the *x*-axis, *y*-axis, and *z*-axis are illustrated. 

From the derivation of the kinematics analysis based on the information in Table 2 [15,16], where $l_{S}$ and $l_{E}$ are defined as the lengths of the first sub-section and the last sub-section, the wiring hole length is h, and the distance between two consecutive wiring holes is g. Accordingly, $\theta_{RL}$ as the right–left bending angle can be calculated with $\theta_{1},\theta_{3},\ldots ,\theta_{13}$, and $\theta_{UD}$ as the up–down bending angle with $\theta_{2},\theta_{4},\ldots \theta_{14}$. The homogeneous transformation matrix (HTM) adopted is shown in Equation (1).
(1)$$A_{n}=\left[\begin{array}{cccc} c\theta_{n} & -s\theta_{n}c\alpha_{n} & s\theta_{n}s\alpha_{n} & a_{n}c\theta_{n}\\ s\theta_{n} & c\theta_{n}c\alpha_{n} & -c\theta_{n}s\alpha_{n} & a_{n}s\theta_{n}\\ 0 & s\alpha_{n} & c\alpha_{n} & d_{n}\\ 0 & 0 & 0 & 1 \end{array}\right]$$
where $s\theta_{n}=sin\theta_{n}$, $c\theta_{n}=cos\theta_{n}$. Accordingly, the complete HTM can be calculated using Equation (2) where $P_{x}$, $P_{y}$, and $P_{z}$ represent the end position.
(2)$$T{=A}_{1}\times A_{2}\times \cdots \times A_{n}=\left[\begin{array}{cccc} n_{x} & o_{x} & a_{x} & P_{x}\\ n_{y} & o_{y} & a_{y} & P_{y}\\ n_{z} & o_{z} & a_{z} & P_{z}\\ 0 & 0 & 0 & 1 \end{array}\right].$$

To further investigate the working space, the $\theta_{RL}$ can be ±210° and $\theta_{UD}$ can be ±240° since there are seven 30° gaps for the right–left bending and eight 30° gaps for the up–down bending angle, as shown in Figure 3.

Moreover, a CMOS sensor/camera (diameter < 2 mm) was equipped to the tip of the bending section within the distal head with a diameter of <4 mm, as shown in Figure 5. 

### 2.2. Control Scheme

As shown in Figure 6, the basic control scheme of the proposed miniature serpentine robotic bronchoscopy is composed of a central controller to coordinate the movements of this robotic bronchoscopy and an actuation module to activate the wires with a pull/release motion that can control the bending angle and the bending direction of the bending section at the front portion of this robotic bronchoscopy based on the commands from the central controller. For further applications, this actuation module will be mounted on a collaborative robot arm where the proper orientation can be provided to adapt to the environment. Moreover, a visual information-processing module will be introduced to process the visual information captured by the integrated CMOS sensor/camera at the tip of the bending section. The visual information-processing module can calculate the central position of an area of interest in an image based on the image-processing algorithm. This centering capability will be utilized to automatically control the tip of the bending section pointing to the center of an area of interest. In the following Section 2.3 and Section 2.4, the proposed path planning and navigation scheme will be specified.

### 2.3. Backward Path Planning

As described above, efficient path planning for navigating the robotic bronchoscopy inside the bronchial tree is necessary. Several previous researchers [8,9,10,11,12,13,14] have addressed this issue. In this paper, a backward path planning algorithm is proposed, followed by a forward navigation algorithm. With these paired backward-path-planning and forward-navigation algorithms, it is not required to calculate the accurate position/orientation of the tip of the bending section of the proposed miniature serpentine robotic bronchoscopy; instead, an event planning and navigation scheme conducts the movement of the robotic bronchoscopy. 

The structure/topology of the bronchial tree of a human body does not dramatically change in a short period of time, that is, certain bronchi do not go missing or disappear suddenly. Thus, the proposed backward path planning scheme adopts this invariance of the structure/topology of the bronchial tree as the key concept to encode the invariant structure/topology into discrete events. The basic idea of this path planning is to start planning from the destination and then move back toward the origin/start and record the path that is followed, then reverse the record of the movement path. Finally, the path planning from the origin/start to the destination is achieved. The block diagram of this backward path planning scheme is illustrated in Figure 7.

In Figure 7, the *Destination* is defined as the nearest bronchi to the target of the robotic bronchoscopy. The *Origin* is usually the oral cavity. It shall be noted that before starting path planning, medical image information such as CT scanned images, MRI images, and even a 3D model of the bronchial tree is made available. In this paper, the method of reconstructing a 3D model of the bronchial tree from medical image information is not addressed. 

Conventional path planning is specified as finding the appropriate path from the *Origin* to the *Destination*. Most path-planning algorithms, such as those in [17,18], first assign the destination and then start path searching from the origin. These algorithms accordingly determine a feasible path from the search results. This kind of path planning with a forward-searching process takes longer. This paper proposed a backward scheme for path planning by beginning at the *Destination*. The process for this backward path planning is shown in Figure 7. From the *Destination*, at *Initialization*, *Event Index i* is set to 1 and the *Wall Status* is set to “*Not detected*”. An *Event* is defined as a situation that the tip of the robotic bronchoscopy will encounter during its trip from the *Origin* to the *Destination*. In the bronchoscopy use case, the *Event* is further specified as Equation (3), where *Node* refers to a bifurcation in the bronchial tree and the *Decision* represents what decision is made at such a *Node*. Thus, *Event* means that at such a *Node,* a predetermined *Decision* shall be made to trigger this *Event*. The *Decision* can be *Right* or *Left* in the simplified bronchial tree, as shown in Figure 8, which means the tip of the robotic bronchoscopy shall move right or left at such a *Node* to trigger that associated *Event*.
(3)$${Event}_{i}={Node}_{i}+Decision$$.

*Wall Status* represents whether there is a *Wall* detected in the image captured by the robotic bronchoscopy. *Wall* signifies that there is a line existing between two bronchi in the captured image, as shown in Figure 8. After *Initialization*, a sliced image at the *Destination* is loaded. *Node Determination* will determine whether there is a *Node* or not based on the detection of a *Wall*. If a *Wall* is detected by image processing to identify a line existing between two bronchi, then a *Node* at a bifurcation point in the bronchial tree is found. In this *Node Determination*, a change in *Wall Status* is critical information to avoid duplication of the same *Node*. Therefore, based on the result from *Node Determination*, a new *Event* could be added. The Event is recorded, and 1 is added to *Event Index i* as Equation (4) shows.
(4)$${TEvent}_{i}={TNode}_{i}+Decision\left(RorL\right).\;i=i+1$$

Then, the tip of the robotic bronchoscopy can move backward toward the *Origin* with one step backward after image loading. The *Decision* at this newly determined *Event* can be derived because, after moving back to the *Origin,* the virtual path the tip has passed can be recorded. A series of *Decisions* would be recorded accordingly, which represents a series of *TEvents* that are defined accordingly. When the *Origin* is reached virtually by the tip of the robotic bronchoscopy, a series of *TEvents* in reverse order from the *Destination* to *Origin* is specified by our proposed path-planning process. To reverse the backward order, we set *Final Event Index FEI* = current *Event Index i*, then according to Equation (5), Event Path Planning with an incremental index from the *Origin* to the *Destination* is generated.
(5)$$\mathrm{From}\;{j=1\;to\;FEI-1,{Event}_{j}=TEvent}_{FEI-j+1}.$$

At the Navigation stage, the robotic bronchoscopy is controlled to trigger this specified series of *Events* from the *Origin*, and finally, the *Destination* will be reached. 

### 2.4. Forward Navigation

By adopting the proposed backward path planning described in Section 2.3, a series of *Events* with *Nodes* and *Decisions* is derived from the *Origin* to the *Destination*. In Figure 9, the navigation process is demonstrated. The core concept is to adopt image recognition to identify the *Node*, make a *Decision* at this *Node* and mark this *Event* as triggered, and then move forward until the next *Node* is identified. This kind of navigation does not strictly depend on the accurate position of the tip of the robotic bronchoscopy compared to conventional navigation with a robotic endoscope. Furthermore, this navigation depends on ensuring a series of *Events* is triggered in order. As shown in Figure 9a, following *Events* 1~3 and making the corresponding *Decisions* 1~3 by identifying *Nodes* 1~3, the *Destination* can be reached. In Figure 9b, the complete navigation process is depicted. 

As shown in Figure 9b, navigation starts from the *Origin* via several functional processing modules and guides the robotic bronchoscopy to move forward to the *Destination*. During the entire navigation process, *Image Capture* means capturing real-time images from the CMOS sensor/camera at the tip of the robotic bronchoscopy. Because there is an orientation indicator located at the tip, the orientation of the tip can be derived at the *Pose & Orientation Calculation* stage. This orientation information will be used in the calculation of the desired bending angle when a *Decision* (Right/Left) is made. At the *Node Recognition* stage, if a *Wall,* as defined in Section 2.3, is identified by image processing, then the *Depth Calculation* will determine the *Depth* from the tip to the *Wall* and pass the depth information on to the next stage *Event-triggered Navigation*. If no Wall is identified, the controller will control the tip of the robotic bronchoscopy to point to the center of the captured image and then keep moving forward. This *Centralization* can be achieved by visual servo control as described in our previous research [16]. 

At the *Event-triggered Navigation* stage, a detailed procedure takes place, as is shown in Figure 10. Based on the pose, orientation, depth information, and event-triggered history, the bending section of the robotic bronchoscopy is controlled to point toward the identified *Wall,* as shown in Figure 11, via an estimation. During this process, if the calculated depth is small enough, then the bending section is controlled to move toward the center of the desired branch of the bronchial tree according to the planned *Event + Node + Decision* as shown in Figure 12. 

During this navigation, it is necessary to record all the *Events* being triggered and compare them with the pre-determined *Event* planning. Accordingly, following the ordered *Event* planning, the robotic bronchoscopy can move toward the desired *Destination*.

In the following Section 2.5, an extra module is proposed with a virtual impedance force control scheme to avoid the tip of the robotic bronchoscopy making forceful contact with the bronchial wall. In contrast to ref. [19], in this paper, two kinds of virtual forces are generated to maintain the tip in a central position inside the bronchi. 

### 2.5. Virtual Force for Centering 

This paper proposes a module with a virtual impedance force control scheme in order to avoid unnecessary contact during operation. Figure 13 demonstrates this scheme. Two virtual forces are defined as *F_A_*, representing the attraction force that attracts the tip toward the center, and *F_R_*, representing the repulsion force that pushes the tip away from the bronchial wall. The directions of *F_A_* and *F_R_* point to the center along with the inverse direction of *P*, calculated as Equation (6).
(6)$$F_{A}=-\frac{M_{A}\ddot{P}+B_{A}\dot{P}+K_{A}P}{{D_{A}}^{2}}\;F_{R}=-\frac{M_{R}\ddot{P}+B_{R}\dot{P}+K_{R}P}{{D_{R}}^{2}}$$
where *P* is defined as a vector pointing to the tip position from the center; *M_A_*, *B_A_*, and *K_A_* are the impedance coefficients for attraction; *M_R_*, *B_R_*, and *K_R_* are the impedance coefficients for repulsion; *D_A_* is the distance between tip position *P* and the center; *D_R_* is the distance between tip position *P* and the virtual bronchial wall by setting the radius *R_BW_* from the center. Apparently, based on Equation (6), when *P* is approaching *R_BW_*, *D_A_* is close to *R_BW_*, *D_R_* is then very short, and the repulsion force *F_R_* will increase and strongly push the tip back to the center. When *P* is close to the center, *D_A_* is close to zero, *D_R_* is near *R_BW_*, and the attraction force *F_A_* will attract the tip to the center. By selecting the impedance coefficients (*M_A_*, *B_A_*, *K_A_*) and (*M_R_*, *B_R_*, *K_R_*), the desired attraction and repulsion forces can be generated. In our implementation, we will integrate the impedance control scheme into the tip of our robotic bronchoscopy system.

## 3. System Implementation and Results

### 3.1. Implementation

Based on the mechanical design described in Section 2.1, consecutive sub-sections of the bending section of the robotic bronchoscopy are manufactured by precise laser engraving of one piece of stainless-steel tube, of which the diameter can be selected to satisfy the size of the CMOS sensor/camera adopted. Four metal wires pass through the holes of each sub-section to actuate bending with any desired direction and angle. Figure 14 shows pictures of the bending sections with outer diameters of 3.36 mm and 4.0 mm, respectively, as well as the 3D orientation of this bending section. A CMOS sensor/camera (OV6948 with an optical size of 1/36″, 200 × 200 pixels, and 30 fps) is integrated into the distal head at the tip of the bending section. With reference to Figure 1, it displays the assembly of this robotic bronchoscopy. It shall be noted that SUS304 and SUS316L are applied as the materials of mechanical components to satisfy the FDA requirements for materials. Moreover, the overall manufacturing and assembly process will be GMP-certified. In addition, a gravitational indicator is added to our bronchoscopy implementation to indicate the horizontal level for determining the right or left side.

As per the implementation of the control scheme, the system diagram is shown in Figure 15. An eMIO controller is adopted as a central control coordinator to coordinate the movements of the robot arm and the proposed robotic bronchoscopy, primarily based on the image information from the CMOS sensor/camera and our proposed path planning and navigation algorithms. An SJ605 Collaborative Robot developed by ITRI is adopted in this system. Two RE30 DC motors and correlated drivers and Ethercat controllers are utilized to actuate the wires to control the bending section of the robotic bronchoscopy. Visual servo functionality is implemented to guide the tip of the bending section to aim at the desired position. In Section 3.2, simulated results of the proposed backward path planning and forward navigation are described.

### 3.2. Results

To navigate the robotic bronchoscopy when moving inside the bronchial tree by following the pre-determined path planning with planned and ordered *Events*, an algorithm of backward path planning was described in Section 2.3. Referring to Figure 7, which shows the proposed backward path planning, and Figure 8, which shows the simplified bronchial tree, as shown in Figure 16, an initialization image at *Destination D* is loaded, and at point ① according to the *Node Determination*, *Wall Detection* is *NO*. This leads to *No New Node Added* as shown in Figure 16. Based on the proposed backward path planning, the virtual tip of the robotic bronchoscopy is controlled to move backward from the *Destination* to the *Origin* while the corresponding medical images are loaded along with the backward moving path. Thus, at point ②, the same status is concluded: *No New Node Added*. At point ③, *Wall Detection* is *YES* and *Wall Status* is changed from *Not detected* to *Detected*. In addition, from the image history and 3D model of the bronchial tree reconstructed from medical images, a Right-Hole will appear on the right side once the tip has moved backward slightly from point ② to point ③, then a *New Event* with *New Node* and *Decision* (${TEvent}_{1}={TNode}_{1}+Decision\left(L\right)$) is added, as shown in Figure 16. In the following steps, the *Decision* corresponding to an *Event* is specified by the same scheme. At point ④, the *Wall* is still detected but *Wall Status* is not changed, and the planning result is *No New Node Added*. Meanwhile, at point ⑤, there is no *Wall* detected since the tip of the bronchoscopy is located far from the previous *Wall*. The planning result is *No New Node Added*. Meanwhile, at point ⑥, *Wall Detection* is *YES* and *Wall Status* is changed from *Not detected* to *Detected*, and then a *New Event* with a *New Node* and *Decision* (${TEvent}_{2}={TNode}_{2}+Decision\left(L\right))$ is added. At point ⑦, the status is the same as at point ⑤, and *No New Node* is added. At point ⑧, accordingly, *Wall Detection* is *YES* and *Wall Status* is changed from *Not detected* to *Detected*, and then a *New Event* with a *New Node* and *Decision* (${TEvent}_{3}={TNode}_{3}+Decision\left(R\right))$ is added. At point ⑨, the planning result is *No New Node Added*. Finally, at point ①, the Origin, the *Final Event Index FEI* is 4, and by applying Equation (5), a series of planned and ordered Events are listed, as shown in Table 3.

At the navigation stage, following the navigation algorithm described in Section 2.4 and as shown in Figure 9, by adopting image recognition to identify the *Node*, making a *Decision* at this *Node*, marking and recording that this *Event* has been triggered, and moving forward until the next *Node* is identified, the *Destination* will be reached. In Figure 17, the results of an experiment in the bronchial tree phantom with our proposed algorithms are illustrated. The correctness of image recognition can navigate the robotics bronchoscopy to move along the correct path. In Figure 17a, image recognition is illustrated as yellow and blue circles. During the insertion of the robotic bronchoscopy along the bronchi, a bifurcation is then detected, illustrated as two blue circles in Figure 17b. According to the first *Event* definition, *Right Decision* is selected as indicated in Figure 17b. Meanwhile, in Figure 17c,d, the *Decisions* of the second and third *Events* are *Left* and *Left*, respectively. This experiment demonstrates the feasibility of the proposed algorithms. 

## 4. Discussion and Conclusions

In this paper, a review of MSC therapy has been addressed. Researchers have preliminarily demonstrated the potential effectiveness. Therefore, a miniature serpentine robotic bronchoscopy is proposed to deliver MSCs as close to the target as possible. Furthermore, a corresponding mechanical design and control scheme has been also discussed. In addition, a backward path planning and forward navigation algorithm is described to efficiently perform path planning from the oral cavity to the deep bronchi and navigate the robotic bronchoscopy to move correctly to the planned destination. 

In the proposed path planning and navigation algorithms, the correctness of image processing to identify a Wall and map the identified image to the planned *Event* is significant. Furthermore, a major point of difference compared to other navigation algorithms or solutions is that the accurate position of the tip of the robotic bronchoscopy is not required during the entire navigation in the proposed navigation scheme in this paper. The only information required is a correct map and record of the planned *Events* that are triggered. In addition, making the related appropriate *Decision* at the correct *Node* is also significant.

The implementation of this robotic bronchoscopy with path planning and navigation has demonstrated the feasibility of this solution. More simulations and experiments will be conducted to further verify the efficiency of the proposed path planning and navigation mechanism. In addition, a force sensor is under design and development, which will be attached to this proposed robotic bronchoscopy to provide extra sensory information such as haptic/force feedback. The provision of the haptic/force information supports the system in reducing the probability of tissue damage due to any applied force. With this haptic/force information, this robotic bronchoscopy can perform more sophisticated operations within limited space inside the human body. Moreover, an MSC cell sprayer is under development to spray MSCs uniformly onto the target with the working channel provided via the proposed miniature serpentine robotic bronchoscopy. This proposed robotic bronchoscopy can be a platform for carrying various tools with their own specific purposes. Moreover, with multiple miniature serpentine robotic endoscopes, more complicated and dexterous operation procedures could be performed. 

## 5. Patents

USPTO Patent Application No. 18/071,450.

## Figures and Tables

**Figure 1 micromachines-14-00969-f001:**
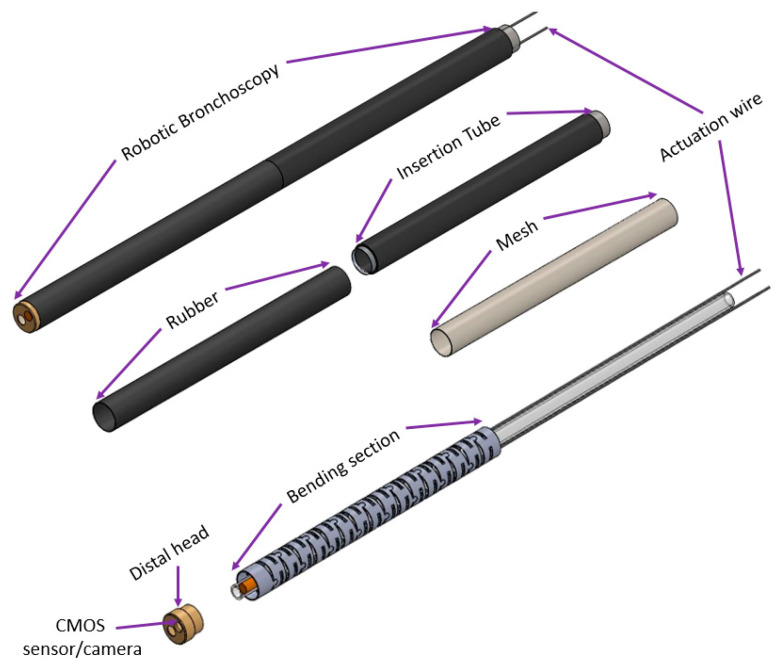
The mechanical structure of this proposed robotic bronchoscopy.

**Figure 2 micromachines-14-00969-f002:**
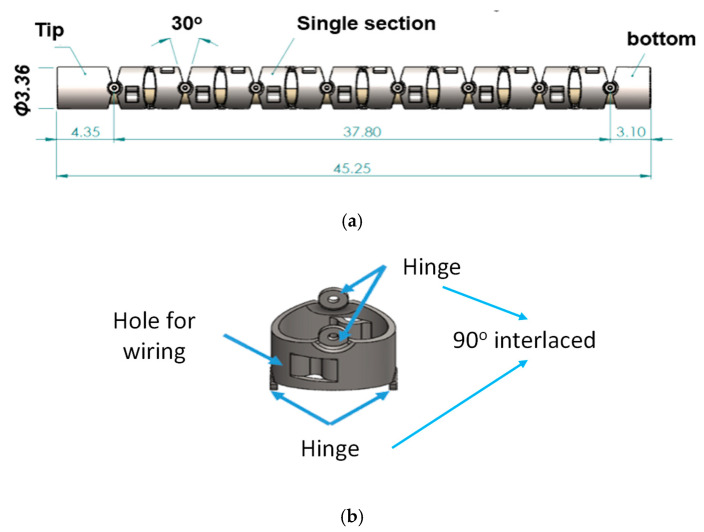
(**a**) 30° gap and (**b**) 90° interlaced placement of hinges between two consecutive sub-sections.

**Figure 3 micromachines-14-00969-f003:**
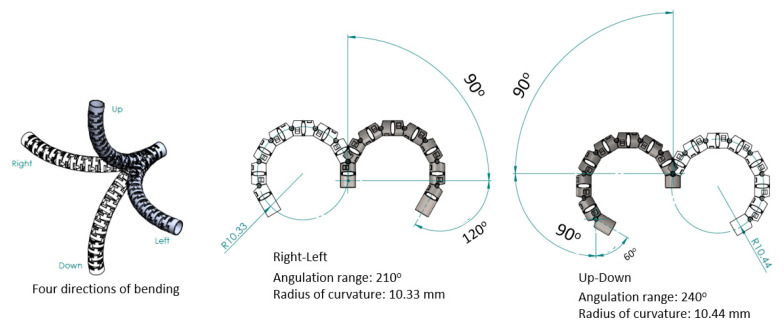
Bending angles of bending section composed of eight sets of two sub-sections.

**Figure 4 micromachines-14-00969-f004:**
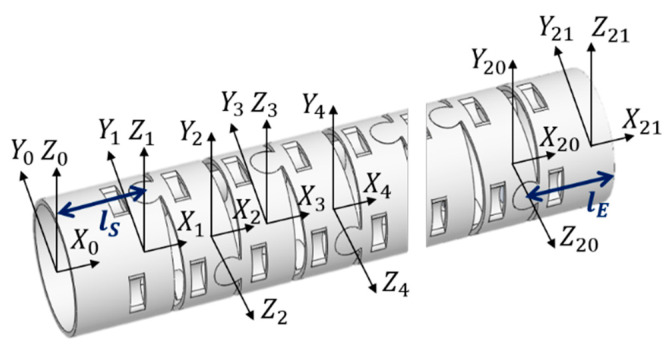
Coordinate assignment for each sub-section [15,16].

**Figure 5 micromachines-14-00969-f005:**
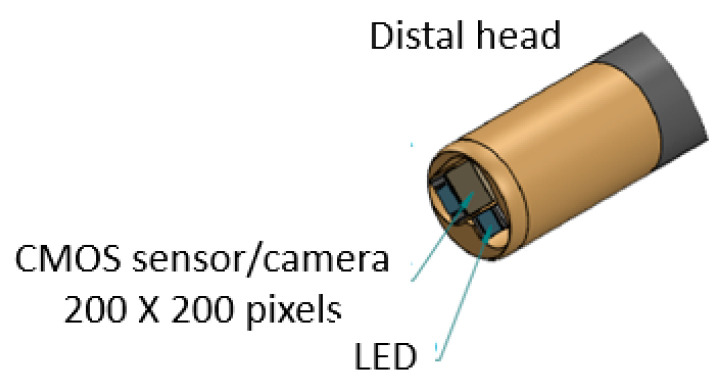
CMOS sensor/camera at the tip of the bending section within the distal head.

**Figure 6 micromachines-14-00969-f006:**
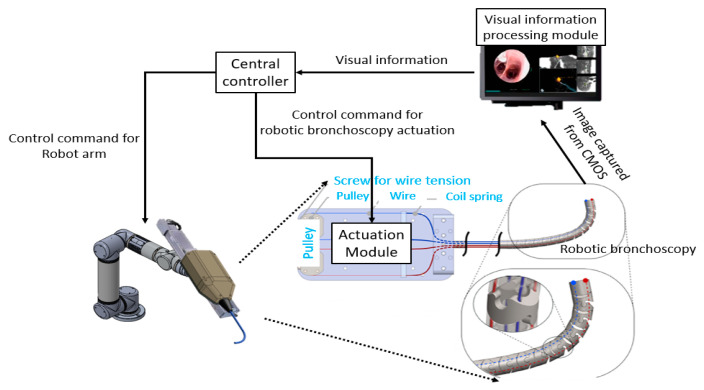
Basic control scheme of the proposed robotic bronchoscopy.

**Figure 7 micromachines-14-00969-f007:**
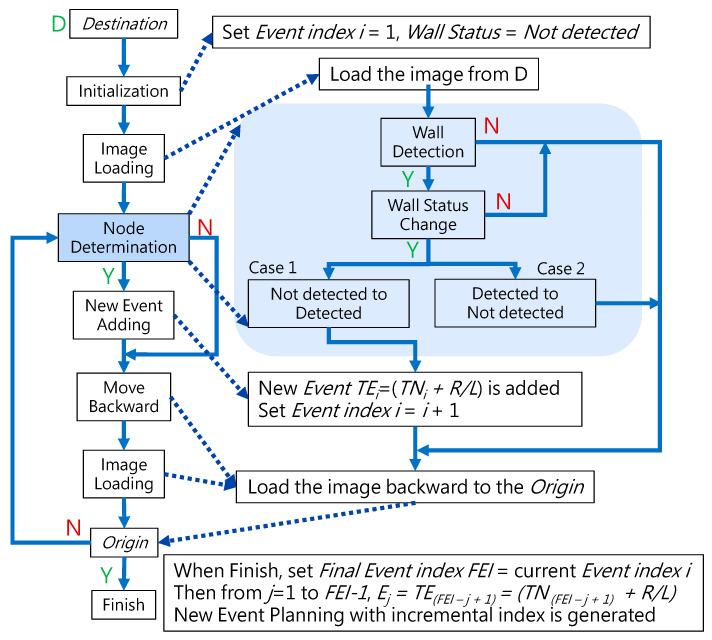
Block diagram of the proposed backward path planning.

**Figure 8 micromachines-14-00969-f008:**
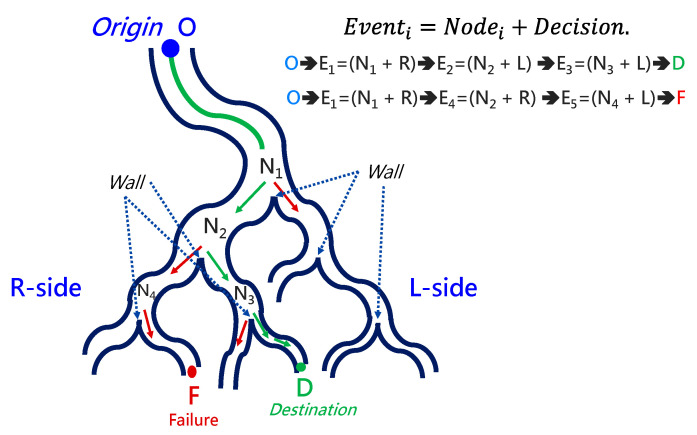
Simplified bronchial tree showing *Event*, *Node*, *Wall*, and *Decision*.

**Figure 9 micromachines-14-00969-f009:**
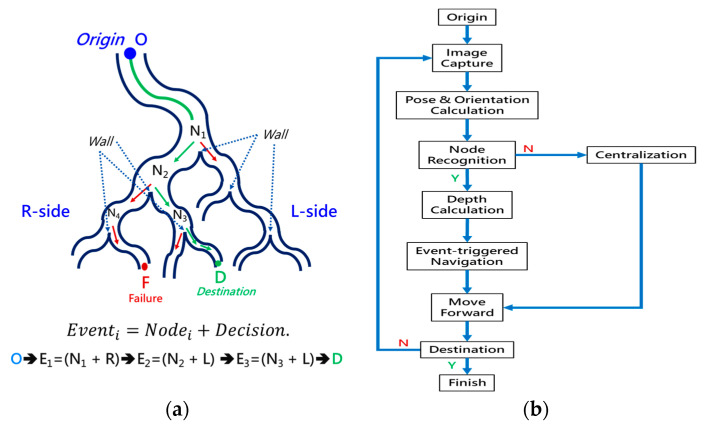
Navigation process: (**a**) A simplified bronchial tree with *Event* planning from the *Origin* to the *Destination*; (**b**) the detailed navigation process block diagram.

**Figure 10 micromachines-14-00969-f010:**
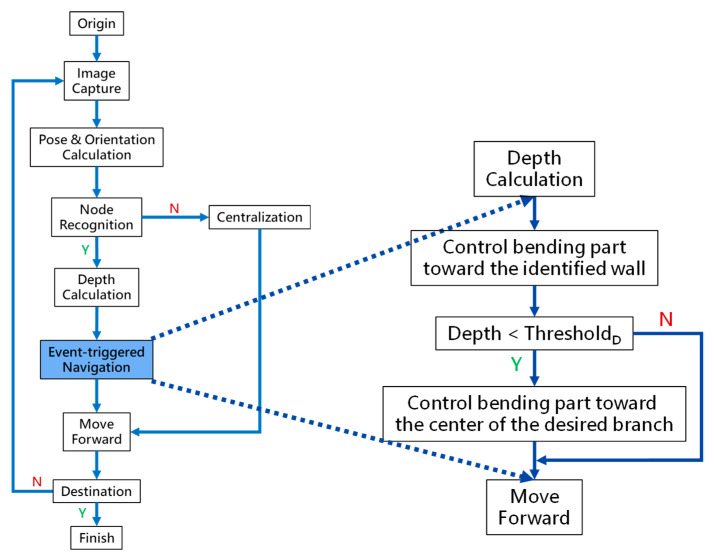
Detailed procedure of Event-triggered Navigation.

**Figure 11 micromachines-14-00969-f011:**
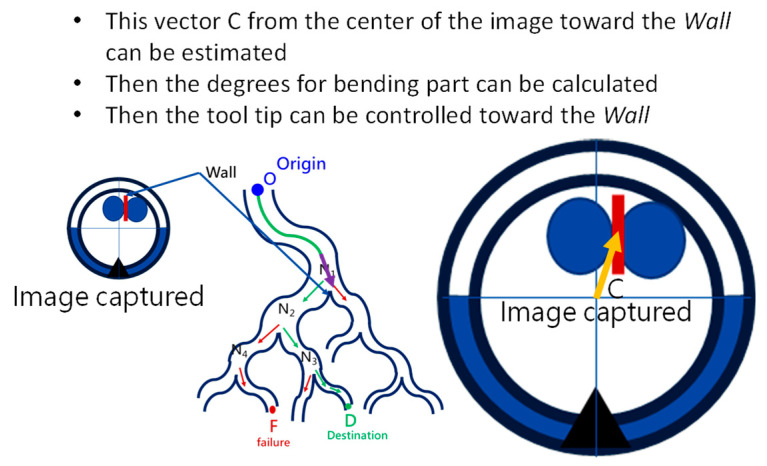
Bending section of the robotic bronchoscopy controlled to point toward the identified *Wall*.

**Figure 12 micromachines-14-00969-f012:**
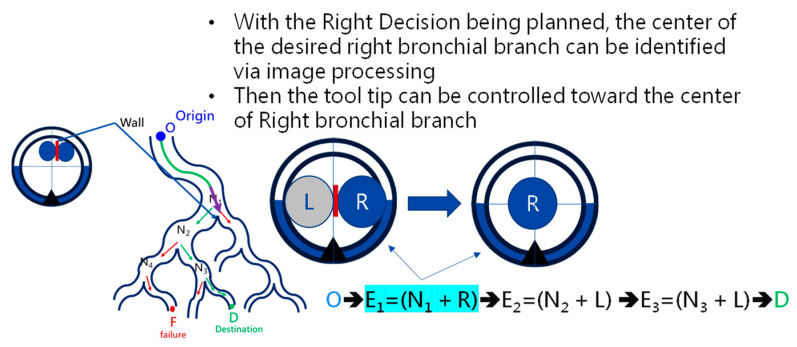
Bending section is controlled to move toward the center of the desired branch of the bronchial tree.

**Figure 13 micromachines-14-00969-f013:**
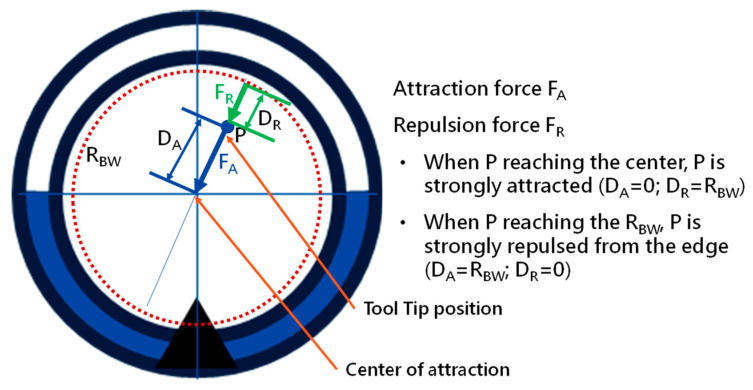
Virtual impedance force control scheme.

**Figure 14 micromachines-14-00969-f014:**
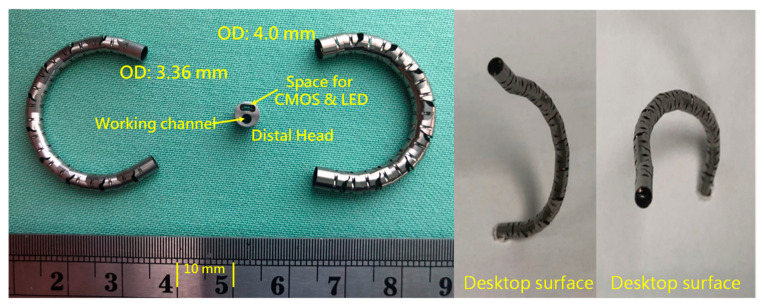
Pictures of the bending section of the robotic bronchoscopy.

**Figure 15 micromachines-14-00969-f015:**
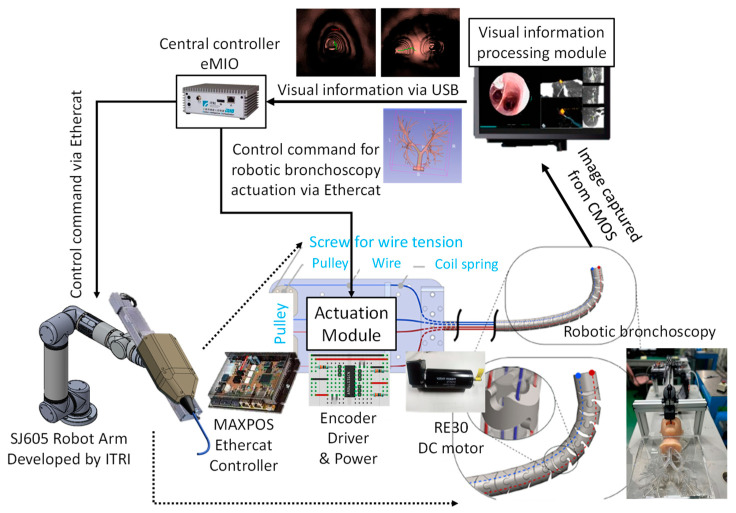
System implementation diagram of the proposed robotic bronchoscopy.

**Figure 16 micromachines-14-00969-f016:**
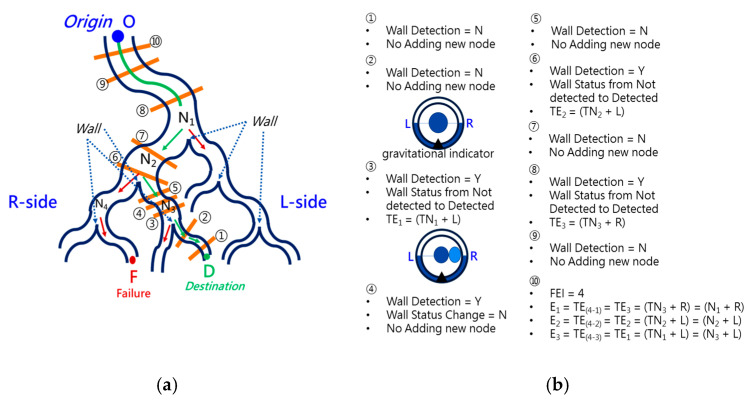
Simulated backward path planning: (**a**) Simplified bronchial tree; (**b**) corresponding path planning at points ① to ⑩.

**Figure 17 micromachines-14-00969-f017:**
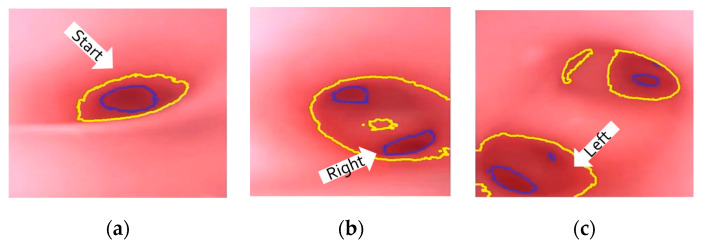
Experiment results in bronchial tree phantom with the proposed algorithms: (**a**) yellow and blue circles shows the results of image recognition at the start; (**b**) Two blue circles show the bifurcation and *Right Decision* is indicated; (**c**) *Left Decision* at *Event_2_* is indicated; (**d**) *Left Decision* at *Event_3_* is indicated.

**Table 1 micromachines-14-00969-t001:** Basic design guidelines and requirements of the proposed robotic bronchoscopy.

Item	Description	Guidelines/Requirements
1.	Bending section at the front portion of the robotic bronchoscopy	Maximal bending angle >90°, larger is better. Four movable directions—Up-Down; Right-Left Radius of curvature/bending section is as less as possible.
2.	Outer diameter of the bending section	Smaller is better. Currently, <4 mm is feasible.
3.	Real-time and in-situ visual information	Visual information is captured by a CMOS sensor/camera. Smaller CMOS sensor/camera is better. Higher resolution is better. Higher frame rate is better.

**Table 2 micromachines-14-00969-t002:** The D-H table of the bending section [15,16].

Joint	*θ*	*d*	*a*	α
1	0	0	l_S_	0°
2	θ_1_	0	g + h	90°
3	θ_2_	0	g + h	−90°
4	θ_3_	0	g + h	90°

14	θ_13_	0	g + h	90°
15	θ_14_	0	l_E_	−90°

**Table 3 micromachines-14-00969-t003:** The simulated results of backward path planning based on Figure 16.

*Event*	*TEvent*	*TNode*	*Node*	*Decision*	*FEI = 4*
*Event_1_*	*TEvent_(4-1)_ = TEvent_3_*	*TNode_3_*	*Node_1_*	*R*	
*Event_2_*	*TEvent_(4-2)_ = TEvent_2_*	*TNode_2_*	*Node_2_*	*L*	
*Event_3_*	*TEvent_(4-3)_ = TEvent_2_*	*TNode_1_*	*Node_3_*	*L*	

## Data Availability

Data sharing is not applicable to this article.

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
