# Peer review of "Path Planning and Navigation of Miniature Serpentine Robot for Bronchoscopy Application"

_micromachines, 2023, doi:10.3390/mi14050969_

Round 1

Reviewer 1 Report

The paper is well organized and written. But the experiment section seems to be a little weak, no data-related results were demonstrated and discussed. It should be better if you have more experimental results with the proposed method.

Author Response

Reply: Thanks for the valuable comment. Currently, simulation has been performed as shown in the manuscript. The algorithm for backward path planning can plan the path for the robotic bronchoscopy moving from the origin to the destination. In the revised manuscript, results of imaging processing for recognizing the target bronchial branch are added for further demonstrating the feasibility of the algorithms proposed in this paper in Figure 17. This experiments are performed in bronchial tree phantom with our proposed algorithms. The correctness of image recognition can navigate the robotics bronchoscopy to move along with the correct path planning.     

Reviewer 2 Report

The authors presented a miniature serpentine robotic bronchoscopy to deliver MSCs as close as possible to the target, with a discussion of the corresponding mechanical design and control scheme. A backward path planning and forward navigation algorithm is described to efficiently perform path planning from the oral cavity to the deep bronchi and navigate the robotic bronchoscopy to move correctly to the planned destination. The following issues need to be addressed before being accepted. 

1. Representative virtual force during a procedure needs to be presented in the results section, if the virtual force is quantitative. 

2. How effective of using the designed robot to perform the delivery of MSC, and how is it comparable to current studies?

3. Will adding the backward path planning and forward navigation algorithm and virtual force affect the speed/time of the procedure?

Moderate editing of English language

Author Response

1. Representative virtual force during a procedure needs to be presented in the results section, if the virtual force is quantitative. 

Reply: Thanks for the valuable comments. In this paper, a virtual force is introduced for restricting the movement of the tip of the proposed robotic bronchoscopy to heavily contact with the bronchial wall. In this paper, the main theme is to discuss the feasibility of the proposed algorithms of path planning and navigation. The introduced virtual force mechanism is an extra feature but shall be described for completeness of the design of the proposed robotic bronchoscopy.

It is agreed that there are still many issues related to this virtual force that require more investigation and discussion as reviewer 2 mentioned. For example, (i) how to select the parameters of the impedance M, B, K; (ii) is that possible for various applications/operations to set various combinations of the impedance parameters; or (iii) how sensitive the robotic bronchoscopy shall be (since the working space of the tip of the robotic bronchoscopy is limited in a circle with diameter less than 20 mm and even smaller to 5 mm). The accuracy and resolution of position and bending control of the tip movement will impact the performance. And these issues will be addressed in another paper (in preparation) focusing on force/haptic information/feedback within the robotic bronchoscopy applications.     

2. How effective of using the designed robot to perform the delivery of MSC, and how is it comparable to current studies?

Reply: Thanks for the comments. In this paper, the main focus is on how to design the robotic bronchoscopy and how to perform path planning and navigation for this proposed robotic bronchoscopy. With the proposed design and algorithms, this robotic bronchoscopy can be as a platform for further bronchoscopic operations. The effectiveness of delivering MSC via the working channel designed within this proposed robotic bronchoscopy will be discussed in another paper since a special container of storing the MSCs is under design with another ITRI’s team.

3. Will adding the backward path planning and forward navigation algorithm and virtual force affect the speed/time of the procedure?

Reply: Thanks for the comments. Since we do not implement another path planning and navigation mechanisms with our robotic bronchoscopy this comparison cannot be made at this moment. Currently, the main purpose of this paper is to discuss the feasibility of our proposed design. This suggestion from reviewer 2 will be taken into significant consideration in our future work.

Reviewer 3 Report

This paper presents path planning and navigation of miniature serpentine robot for bronchoscopy application. The authors have applied a miniature serpentine robot to NOTES. To make this possible, they describes the design and control scheme. Overall their method of planning and navigation looks interesting and well behaves in-situ environments. In addition, the quality of the paper looks good and well organized. This paper is worth accepting after double checking English.

Minor check is required to improve the paper quality.

Author Response

This paper presents path planning and navigation of miniature serpentine robot for bronchoscopy application. The authors have applied a miniature serpentine robot to NOTES. To make this possible, they describes the design and control scheme. Overall their method of planning and navigation looks interesting and well behaves in-situ environments. In addition, the quality of the paper looks good and well organized. This paper is worth accepting after double checking English.

Reply: Thanks for the comments.

Round 2

Reviewer 2 Report

The authors have addressed all of my comments.

Minor editing of English language required